# Immunosurveillance and Immunoediting of Lung Cancer: Current Perspectives and Challenges

**DOI:** 10.3390/ijms21020597

**Published:** 2020-01-17

**Authors:** Kei Kunimasa, Taichiro Goto

**Affiliations:** 1Department of Thoracic Oncology, Osaka International Cancer Institute, Osaka 541-8567, Japan; keikunimasa@gmail.com; 2Genome Analysis Center, Yamanashi Central Hospital, Yamanashi 400-8506, Japan; 3Lung Cancer and Respiratory Disease Center, Yamanashi Central Hospital, Yamanashi 400-8506, Japan

**Keywords:** non-small cell lung cancer, immunoedition, immune checkpoint inhibitors, immunotherapy

## Abstract

The immune system plays a dual role in tumor evolution—it can identify and control nascent tumor cells in a process called immunosurveillance and can promote tumor progression through immunosuppression via various mechanisms. Thus, bilateral host-protective and tumor-promoting actions of immunity are integrated as cancer immunoediting. In this decade, immune checkpoint inhibitors, specifically programmed cell death 1 (PD-1) pathway inhibitors, have changed the treatment paradigm of advanced non-small cell lung cancer (NSCLC). These agents are approved for the treatment of patients with NSCLC and demonstrate impressive clinical activity and durable responses in some patients. However, for many NSCLC patients, the efficacy of immune checkpoint inhibitors is limited. To optimize the full utility of the immune system for eradicating cancer, a broader understanding of cancer immunosurveillance and immunoediting is essential. In this review, we discuss the fundamental knowledge of the phenomena and provide an overview of the next-generation immunotherapies in the pipeline.

## 1. Introduction

In 1891, William Coley noticed several clinical cases of spontaneous regression and disappearance of tumors in patients following infection [1,2]. He conceived the idea of injecting bacteria into his cancer patients, which showed promising results and became known as Coley’s toxin. In 1909, Paul Ehrlich postulated the concept of cancer immunosurveillance wherein subclinical tumor cells are continuously and spontaneously eradicated by the activated immune system [3]. This concept is now clearly established—primarily through demonstrations of increased cancer incidence in immunodeficient mice and humans [4]. However, cancer develops despite this immunosurveillance. The paradigm of cancer immunoediting reconciles the mechanisms of tumor surveillance with the presence of clinically evident cancers. Tumors derived from mutagenesis in immunodeficient mice are more immunogenic compared to those from immunocompetent mice [4,5]. This phenomenon indicates that tumors develop differently in the presence of immune system. Specifically, the immune system applies a selective pressure to tumor development and growth, and tumors are forced to edit the surrounding immune system, which is known as cancer immunoediting [6,7]. Cancer immunoediting involves three sequential phases within the context of the immune system: the elimination phase, whereby immunogenic tumors are eradicated; the equilibrium phase, whereby tumor cells and immune cells coexist; and, finally, the escape phase, wherein tumor escape from immune control. Clinically detectable cancers often represent this escape phase.

## 2. Cancer-Immunity Cycle

### 2.1. Cancer-Immunity Cycle Outlined

The concept of a “cancer-immunity cycle,” demonstrating the dynamic anti-tumor immune responses of cancer in 7 steps, has been proposed (Figure 1) [8]. Cancer antigens are released by tumor cells during apoptosis or cell death (Step 1). The antigens released are captured by dendritic cells (DCs), which then mature and simultaneously migrate to lymph nodes (Step 2). At the lymph nodes, the DCs present the captured cancer antigen to the major histocompatibility complex class (MHC) I molecule, resulting in the priming of T cells (Step 3). The activated T cells begin the migration to the tumor (Step 4), infiltrate the tumor tissue (Step 5), recognize cancer cells (Step 6), and injure them (Step 7). Steps 3 and 6 serve as immune checkpoints at which the activation of T cells is controlled. From the cancer cells injured by T cells in Step 7, cancer antigens are further released, and the cancer-immunity cycle returns to Step 1 and continues. In many cancer patients, one or more of these steps can be interrupted, resulting in ineffective immune responses to cancer.

### 2.2. CD8-Positive Cytotoxic T Cells

A key step of anti-tumor immune responses is the activation of CD8-positive cytotoxic T cells (CTLs). Cytotoxic T cells, activated by the recognition of cancer-specific antigens presented by antigen-presenting cells (APCs), begin attacking cancer cells [9]. It has been shown that if the immune checkpoint molecules (co-suppression molecules) expressed on immunocompetent cells, such as CTLs are suppressed or the immune co-stimulatory molecules are stimulated, the CTLs are activated, resulting in the reinforcement of anti-tumor immune responses [10].

The action of co-stimulatory and co-suppression molecules involved in T cell reactions and their action on effector T cells and Tregs is explained below (Table 1). In addition, the relationship between co-stimulatory and co-suppression molecules to their ligands is classified by the priming (naïve T cells are activated by antigen presentation by APCs and proliferate; Figure 2) and effector phases (the activated and proliferating cancer-specific T cells gather in the cancer cells and attack; Figure 3).

### 2.3. CD28 Family: Cytotoxic T-lymphocyte Associated Protein 4 and Programmed Cell Death 1

Cytotoxic T-lymphocyte associated protein 4 (CTLA-4) is a co-suppression molecule not expressed on steady-state T cells, but induced in a dynamic manner following the activation of effector CD4+ or CD8+ T cells [11]. CD80 and CD86 expressed primarily on APCs serve as a ligand for CTLA-4 and CD28 (a representative co-stimulatory molecule) [12]. CD28 activates the effector T cells after recognition of a specific antigen and its activation induces the expression of CTLA-4 that binds to CD80 in a manner competing with CD28 with a 10–100 fold higher binding affinity [12,13]. Co-stimulation by CD28 is thus attenuated, resulting in the suppression of immune responses. The CTLA-4 on Treg causes the attenuation of APC function indirectly by emitting a reverse signal to CD80 and CD86 or effector T cell function, directly [14]. The development of systemic autoimmune diseases and death two months thereafter of CTLA-4 knock-out mice also supports its immunosuppressive activity [15].

Programmed cell death 1 (PD-1) is a co-suppression molecule expressed dynamically following the activation of effector T cells [16]. Its representative ligand is PD-L1, expressed primarily on APCs and tumor cells. The expression of PD-L1 on the tumor is induced by IFN-γ released from activated effector T cells following antigen recognition via T cell antigen receptor (TCR) [17]. In the tumor tissue, primarily CD8 positive cytotoxic lymphocytes having undergone activation show high levels of PD-1 expression. If PD-L1 binds to these lymphocytes, tyrosine dephosphorylases (SHP-1, SHP-2, etc.) gather into the intracellular part of the immune checkpoint molecules and inhibit tyrosine phosphorylation of ZAP70, resulting in T cell dysfunction and the induction of apoptosis thereby suppressing excessive anti-tumor immune responses [18]. Cancer antigen-specific CTLs invading tumor cells are selectively suppressed by PD-1, which is constitutively expressed on Treg, suggesting its role in Treg differentiation and function [19]. Supporting this, studies have shown that PD-1 knock-out mice develop diverse autoimmune diseases (e.g., nephritis, arthritis, and myocarditis) spontaneously [20,21].

### 2.4. TNF Receptor Superfamily: OX-40 and Glucocorticoid-Induced TNFR-Related (GITR) Gene

OX-40 is a co-stimulatory molecule; its ligand is OX-40L, which is induced primarily following activation of T cells and APCs [22]. The stimulatory signals of OX-40 promote activation, survival, and proliferation of effector cells and the production of cytokines [23]. OX-40 is constitutively expressed on Treg and involved in the formation and function of Treg. In many animal models, the anti-OX-40 agonistic antibody has been reported to manifest potent anti-tumor activity by inducing the activation of effector T cells and attenuation of Treg function [23].

GITR is expressed on effector T cells at low levels, serving as a co-stimulatory molecule whose expression increases following effector T cell activation [22]. Its ligand, GITRL, is expressed constitutively on APCs and endothelial cells [22]. GITR signals promote the activation and proliferation of effector T cells and suppression of apoptosis. GITR is constitutively expressed on Treg, and GITR signals cause the attenuation of Treg’s suppressive activity [24]. In many animal models, the anti-GITR agonistic antibody is known to manifest anti-tumor activity by causing the activation of effector T cells and attenuation of Treg’s suppressive activity [24].

## 3. Tumors Escape Immunosurveillance through Immunoediting

As revealed by the presence of occult and clinically imperceptible tumors, tumor cells that successfully evaded immunosurveillance can enter tumor dormancy (the equilibrium phase) [25] (Figure 4). In the equilibrium phase, the adaptive immune system prevents tumor invasion and outgrowth and sculpts tumor immunogenicity. However, as a consequence of constant immune selection imposed on genetically unstable tumor cells held in equilibrium, tumor cells can acquire the ability to circumvent immune recognition, thereby avoiding destruction and eventually developing into clinically detectable tumors (the escape phase) [26,27]. The mechanisms controlling tumor escape from immunosurveillance are diverse and include downregulation or loss of expression of MHC I molecules, which are essential for CD8+ cytotoxic T cell recognition [28,29], and increased expression of cytotoxic T cell inhibitory ligands, such as programmed cell death ligand 1 (PD-L1), which suppresses cytotoxic T cell attack [30] (Figure 3). Cytotoxic T cell activation involves multiple steps generated through the immune synapse between tumor and immune cells. The first signal is transduced through the binding of a T cell receptor to its cognate peptide located on surface-expressed MHC molecules. This interaction assigns specificity to the consequence of downstream T cell subsets. Many negative regulatory checkpoints are present to inhibit overactive immune responses and set a limit on T cell activation. In order to escape detection by the immune system, tumor cells upregulate the surface expression of inhibitory molecules, including PD-L1, cytotoxic T-lymphocyte associated protein 4 (CTLA-4), LAG-3, TIM-3, and 4-1BB [31,32,33,34,35]. As a result of the constant exposure to antigens in the tumor microenvironment, intratumoral T cells display a broad spectrum of suppressed and dysfunctional states, called T cell exhaustion [36]. Thus, tumor cells evade immunosurveillance by editing their microenvironment, including immune cells and their neighboring normal cells. The escape phase has been an area of intense investigation in the field of tumor immunology over the past decades. Many studies have demonstrated that tumors in the escape phase evade the immune system through direct and/or indirect mechanisms to aid in their growth and metastases [26]. Focused work is now being undertaken to devise strategies that can target these mechanisms of escape since they represent a means to treat tumors using novel cancer immunotherapies [37,38]. This review provides multiple examples of tumor immunoediting mainly in lung cancer.

## 4. Human Leukocyte Antigen (HLA) Loss and Immune Escape in Lung Cancer Evolution

The MHC, a set of genes that code for cell surface proteins, helps the immune system recognize foreign substances. MHC proteins are inherent in all higher vertebrates. The human MHC complex is synonymous with the human HLA complex. Silencing, downregulation, or loss of HLA alleles inhibits peptide antigen presentation and facilitates tumor cells to escape from immunosurveillance [28,29,39] (Figure 5).

A recent report elucidated loss of heterozygosity (LOH) at the HLA alleles; in particular, loss of HLA-C*08:02 was observed in a resistant lesion treated with tumor-infiltrating lymphocytes composed of cytotoxic T cell clones targeting the KRAS G12D mutation [40]. Since the presence of the HLA-C*08:02 allele is requisite for the presentation of the neoantigen KRAS G12D and tumor recognition by T lymphocytes, its loss was supposed to directly cause immune evasion. In hepatocellular carcinoma, tumors with HLA LOH exhibited significant association with an increased recurrence, indicating that immune escape caused by HLA LOH may accelerate tumor progression [41]. Downregulation of HLA in vulvar intraepithelial neoplasia was also associated with the development of recurrences [42]. Clinically, patients with high evasion capacity tend to have a higher recurrence rate compared to those with low evasion capacity [41,42]. However, the magnitude and significance of the loss of the HLA haplotype has not been systematically evaluated in human cancers, because the polymorphic feature of the HLA locus precludes alignment of sequence reads to the human reference genome and estimation of copy number. To overcome this challenge, McGranahan et al. recently developed a new computational tool using next-generation sequencing data to estimate the allele-specific copy number of the HLA locus; loss of heterozygosity in the human leukocyte antigen (LOH HLA) was found to occur in approximately 40% of non-small-cell lung cancers (NSCLCs) and was associated with high PD-L1 expression [43]. The subclonal frequency of HLA LOH, which occurs in a subset of tumor cells and is located on the branches of evolutional phylogenetic trees, suggests that HLA LOH is usually a late event in tumor phylogeny and that immune microenvironment in the tumor tissue may function as a key selective pressure in shaping branched tumor evolution. In a cohort of primary NSCLC tumors with matched brain metastasis, HLA LOH was detected in 47% of the cases, which occurred subclonally and preferentially at the metastatic brain sites. Thus, HLA LOH is a common feature of NSCLC and facilitates immune escape from immunosurveillance and significant immune editing, leading to subclonal genome evolution.

## 5. Heterogenous Immunoediting in Lung Cancer

Multi-dimensional datasets, such as The Cancer Genome Atlas (TCGA), International Cancer Genome Consortium (ICGC) and Gene Expression Omnibus (GEO), enable the study of relationships between different biological processes, e.g., genome-wide DNA sequencing, DNA methylation, gene expression and copy number change, and the leveraging of multiple data types to draw inferences about biological systems [44,45,46]. Recent TCGA datasets have associated the genomic profile of tumors with tumor immunity, which implicates the neoantigen burden in promoting T cell responses [47] and identifies somatic mutations in relation to immune infiltrates [48]. The TRACERx (TRAcking Cancer Evolution through therapy (Rx)) lung study is a multi-million pound research project taking place since April 2014 [49], which will transform our understanding of NSCLC and take a practical step towards an era of precision medicine. The study will reveal mechanisms of cancer evolution by analyzing the intratumor heterogeneity in lung tumors from approximately 850 patients and tracking its evolutionary trajectory from diagnosis through to relapse. The project enrolled patients to obtain samples of surgically resected NSCLC tumors in stages IA through IIIA for high-depth, multiregion whole-exome sequencing. TRACERx recently demonstrated that the immunological profiles can vary dramatically among different regions of the same early-stage tumor, similar to genomic heterogeneity, which can affect the prognosis of patients [50]. They also assessed immune cell infiltration in 258 samples from 88 treatment-naïve, early-stage, lung cancers by histological analyses and RNA sequencing. Intriguingly, heterogenous immune cell infiltration was associated with genomic heterogeneity; intratumor variations in tumor mutation burden (TMB), may confound some putative biomarkers for precision medicine. To examine the influence of tumor-infiltrating immune cells on tumor evolution, neoantigen burden was compared and analyzed. An analysis of non-neoantigenic, nonsynonymous mutations confirmed a significant reduction in expressed neoantigens—this depletion was limited to tumors with intact HLA alleles and, again, to tumors with a high level of immune cell infiltration, suggesting that these alterations are mediated by immune cells. These data suggest that the immune profile can vary markedly within each tumor and that there is a heterogeneity of immunoediting within the tumor, which affects tumor evolution.

The heterogeneity of immunoediting reflects the mutational burden of lung cancer [51], and the TCR repertoire represents the breadth and strength of the T cell immune response [52]. The enriched TCR repertoire in both tumors and the surrounding normal tissue are composed of thousands of different TCR repertoires. Many of them are present at low frequency and a fraction contains bystander T cell populations [53]. In contrast, the expanded intratumor TCR repertoire is differentially expressed in the tumor compared to the adjacent normal tissue. Intriguingly, the number of expanded intratumor TCR repertoires significantly correlates with the number of nonsynonymous mutations, and the heterogeneity of intratumor TCR repertoire correlates with spatial mutational heterogeneity. Moreover, the number of ubiquitous and regional TCRs correlates with the number of ubiquitous and regional nonsynonymous mutations, respectively. Thus, heterogeneity of the mutation landscape can affect immune landscape heterogeneity.

The immune microenvironment also affects tumor evolution at different metastatic organs [54,55]. Examining the influence of the tumor microenvironment on the metastatic potential of tumor cells has been hindered by the vast diversity of infiltrating immune cells [56] as well as the extensive interactions between tumor cells and neighboring normal cells [57]. Extensive analysis of a patient with metastatic colorectal cancer during an 11-year-old spatiotemporal follow-up revealed that cancer clonal evolution patterns during metastatic progression depend on the immune context at metastatic sites [58]. Moreover, neoantigen depletion was observed in metastatic sites with high levels of tumor-infiltrating T cells [58]. The immunoedited tumor clones were eliminated, whereas the progressive clones were immune privileged in spite of the presence of tumor-infiltrating immune cells [58,59]. Tumor immunoediting influences both intra- and intertumor heterogeneity and sculpts tumor clonal evolution.

## 6. Neoantigen Derived from Mutation

### 6.1. Tumor Antigens

Tumor antigens can be differentiated into five categories: (i) viral antigens; (ii) differentiation antigens; (iii) cancer-germline antigens; (iv) overexpressed antigens; and (v) neoantigens (Figure 6) [60,61,62,63]. As antigens (i)–(iv) can be expressed in normal tissues as well as cancer tissues, these antigens are more likely to promote immunological tolerance and are less likely to bring about effective anti-tumor immune responses [64]. However, neoantigens are supposed to be expressed exclusively in cancer cells due to genomic mutations altering the amino acid sequence. This type of antigen is tumor-specific and can cause an immune response sufficient to kill tumor cells when activated [65,66,67,68].

Thus, tumor-specific antigens originating from cancer cell-specific gene mutation exist as nonself, because they are absent in normal cells and present in cancer cells alone and are called “neoantigens.” The peptide fragment (neoantigen) containing amino acid mutants processed from the mutant protein of cancer cells binds to the MHC molecule and is presented on the cell surface. It can efficiently induce highly reactive specific T cells without undergoing immunotolerance. Gene mutations of cancer cells can be divided into driver mutations (directly associated with cellular oncogenesis) and passenger mutations (not associated with cellular oncogenesis), and both can yield neoantigens [69]. Neoantigens originating from driver mutations of *BRAF, KRAS,* and *p53* have a high potential of being distributed extensively among cancer patients [70]. A vaccine with wide applicability can be developed using these antigens as the target [71]. If the driver mutation is set as the target, immune escape through the loss of antigen from cancer cells is less likely to occur, and higher clinical efficacy is expected. However, it is incorrect to conclude that the entire peptide sequence, including the part of the driver mutation, is presented by APCs and recognized by T cells. In fact, driver mutations containing peptide sequences less likely to be presented as the antigens are found more frequently in cancer cells [72]. In contrast, neoantigens originating from passenger mutation occur at a much higher frequency in cancer cells. However, inter-individual variations in passenger mutations among patients make their detection difficult using conventional technology. Recently, the development of next-generation sequencers enables easier detection through whole-exome analysis [73,74]. In addition, gene fusions are also identified as a source of immunogenic neoantigens which can mediate anticancer immune responses [75,76]. Their computational prediction from DNA or RNA sequencing data necessitates specialized bioinformatics expertise to assemble a computational workflow including the prediction of translated peptide and peptide-HLA binding affinity [73,76]. Thus, personalized cancer immunotherapy may be developed by identifying neoantigen from the gene mutations (mostly passenger mutations), which vary from one case to another and setting a target of treatment at the identified neoantigen.

### 6.2. Anti-Tumor Immune Responses by Neoantigen-Specific T Cells

In recent years, the clinical efficacy of immune checkpoint inhibitors has been demonstrated, motivating the clinical use of these inhibitors in patients with various cancers [77,78]. However, since the response rate to these inhibitors is low, exploration of efficacy-predictive biomarkers identifying patients expected to respond to these inhibitors has been conducted worldwide, and close attention has been paid to the tumor mutational burden as one possible predictor [79,80]. The responses to immune checkpoint inhibitors correlate positively with the total number of gene mutations, and therapies using these inhibitors have been reported to be particularly effective against cancers involving several gene mutations due to extrinsic factors (ultraviolet ray, smoking, etc.) such as malignant melanomas and squamous cell carcinomas of the lungs [81,82]. Furthermore, as an intrinsic factor, it has been reported that patients with cancers involving the accumulation of gene mutations due to deficient mismatch repairs (dMMR) respond more markedly to the anti-PD-1 antibody [83]. This antibody has been used extensively in the clinical practice against many types of solid cancers, which often shows microsatellite instability (MSI), a marker of dMMR [84]. It has been estimated that an increase in the number of gene mutations in cancer cells is associated with an increase in the number of neoantigens formed from such mutations, resulting in an increase in neoantigen-specific T cells, which are activated by immune checkpoint inhibitors and manifest anti-tumor activity [83,85].

Recently, there has been an increase in the number of reports directly suggesting the presence of neoantigen-specific T cells among cancer patients and the clinical significance of the presence of such cells [86]. Zacharakis et al. infused tumor-infiltrating lymphocytes, containing four types of neoantigen-specific T cell clones, into patients with breast cancer and concomitantly administered immune checkpoint inhibitors to these patients and reported that the metastatic foci subsided and the cancer was eradicated completely [87]. Moreover, several studies have also shown that when the antigenic specificity of infused lymphocytes was investigated in cancer patients having survived years following T cell infusion therapy, the neoantigen-recognizing T cell clones were identified with high frequency [88]. Thus, neoantigen-specific T cells are believed to play a central role in anti-tumor immune responses.

In addition, Anagnostou et al. demonstrated that among the patients with NSCLC that responded to immune checkpoint inhibitors, the disappearance of a total of 41 neoantigens (7–18 antigens per case) was noted in the four cases where the disease recurred [52]. The specific T cells against the disappearing neoantigens were detected during the effective period, but decreased during disease progression, suggesting that tumor reduction in response to immune checkpoint inhibitors is mediated by immune responses to neoantigens and that the disappearance of neoantigens serves as one possible mechanism for the development of resistance to therapy [52,89].

The immunosurveillance and immunoediting mechanisms of cancer exist, but the likelihood of the manifestation of these mechanisms can vary depending on the cancer development process or tumor microenvironments of different types of cancer [90,91]. Immunotherapy using immune checkpoint inhibitors can trigger therapy-induced immunoediting (immune reconstruction) in some cancers, possibly leading to the restoration of cancer elimination/equilibrium, while stimulating therapy-induced tumor escape in others [52]. It is, therefore, necessary to understand the mechanisms involved in immunoediting by cancer or therapy-induced immunoediting and devise a strategy capable of preventing the disappearance of cancer antigens and avoiding therapy-induced tumor escape.

## 7. Treg and Tumor Immunity

The immune system distinguishes self from nonself-molecules. Immunotolerance to self-molecules, i.e., a lack of an immune response to molecules, allows the normal self-tissue to escape immune system attacks. Lymphocytes possessing receptors binding strongly to self-antigens undergo apoptosis following negative selection in the thymus, resulting in the elimination before maturation [92]. However, stimulation of peripheral blood mononuclear cells by the tumor-self antigen can induce self-antigen specific CD8+ T cells. Thus, elimination does not occur for all self-reactive T cells. Based on these findings, it was anticipated that self-reactive T cells are regulated by some mechanisms and later that Treg is involved in this mechanism [93,94].

Regulatory T cells (Treg) are a group of CD4 T cells that suppress various immune responses. They express high levels of IL-2 receptor α chain (CD25), CTLA-4, and transcription factor FoxP3 [94]. CD25+CD4+ T cells account for approximately 10% of all CD4+ T cells found in the peripheral blood of normal individuals [95]. Tregs are mostly formed in the thymus as a subset of T cells specialized in immunosuppression. Part of Treg undergoes differentiation from naïve T cells in peripheral lymph tissue (intestinal lymph tissue among others) under certain conditions. It enables self-immunotolerance through the suppression of self-reactive immune responses and thus, plays an important role in the suppression of autoimmune disease [96]. Thymus-derived Foxp3+CD25+CD4+ Treg is indispensable for the introduction and maintenance of peripheral immune self-tolerance. For example, if peripheral T cells of the same strain of normal mice are transferred into T cell-deficient nude mice after removal of CD25+CD4+ T cells (removal of Treg from normal animals), autoimmune diseases such as thyroiditis and gastritis develop spontaneously [97]. The onset of these diseases can be prevented by the replenishment of the Treg. Furthermore, if the aforementioned mice are then inoculated with tumor cells originating from the same strain of mouse, potent anti-tumor immune responses are induced [98]. If the proliferation of allografted antigen-specific Treg is attempted in the same experimental system, immunotolerance for the organ graft can be induced [99].

In case of malignant tumors, Treg suppresses the anti-tumor immune responses, thus contributing to the proliferation of tumor cells. The tumor tissues often have a large number of Treg that are activated after stimulation with the antigen, and the presence of Treg in the tumor serves as a biomarker of a poor prognosis [100].

Tregs manifest immunosuppressive functions via multiple mechanisms, with the most important mechanism being the inhibition of T cell activation through the suppression of APCs [101,102]. Tregs constitutively express CTLA-4, which binds to the CD80/CD86 of APCs, causing their downregulation and the suppression of their T cell activating capability [103,104]. In addition, as Treg expresses high levels of CD25 (IL-2 receptor α chain), it has a high affinity for IL-2. Owing to the transcription factor FoxP3 suppressing the expression of the IL-2 gene, Treg cannot produce IL-2 on its own. Thus, it survives by consuming the IL-2 produced by other cells [105], resulting in the depletion of IL-2 in local environments and making it impossible for effector T cells to become sufficiently activated. Tregs also suppress immune responses by producing immunosuppressive cytokines such as transforming growth factor-β (TGF-β) and IL-10 [100,106].

Therefore, if immunosuppression by Treg is controlled, anti-cancer immunotherapy may become more effective. With this expectation, various attempts at removing Treg by targeting IL-2 and CD25 have been made but have been clinically ineffective. To the best of our knowledge, no agent capable of removing Treg in a reliable manner has been developed to date [107].

In humans, FoxP3 can also be induced by stimulating naïve T cells using antigens, hence, it is not completely Treg-specific. The appropriate definition of a Treg is essential in the development of new drugs. Depending on the combination of CD45RA (a marker of naïve T cell) and FoxP3, FoxP3 positive T cells can be divided into three fractions: (1) CD45RA+FoxP3^low^CD4+ T cell (naïve type Treg); (2) CD45RA-FoxP3^high^CD4+ T cell (effector type Treg); and (3) CD45RA-FoxP3^low^CD4+ T cell (non-Treg without immunosuppressive activity) (Figure 7) [101,108]. Of these fractions, the effector type Treg has the most potent immunosuppressive activity. Most of the Treg invading the tumor tissue is the effector type Treg, while naïve type Treg is abundantly seen in peripheral blood [109]. At present, attempts are being made to develop new means of treatment by which the peripheral naïve type Treg is preserved (by setting a target at the CCR4 selectively expressed on the effector type Treg) and anti-tumor immune responses are activated by setting a target only at the effector type Treg in the tumor tissue [109,110].

## 8. Tumor-Associated Macrophages (TAMs) and Tumor Immunity

Macrophages are cells, found in various tissues of the body, possessing diverse functions such as ontogeny, homeostasis of organisms, tissue repair, and immune responses to pathogen infection. They are conventionally viewed as differentiating from monocytes among leukocytes. In recent years, however, it has been shown in multiple tissues that the macrophages indigenous in the tissue during the fetal period undergo proliferation in that tissue [111,112]. More importantly, the macrophages that invade the tumor stroma are called TAM and stimulate the development, progression, and metastasis of tumors (Figure 8) [113,114,115]. In many tumors, TAM is recruited into microenvironments by CCL2, CSF-1, IL-10, TGF-β, etc. secreted from cancer cells. CSF-1 plays a particularly important role in TAM as it contributes not only to TAM differentiation, but also invasion into tumor tissue [116]. Owing to these activities, the proliferation of Lewis lung carcinoma cells implanted subcutaneously is suppressed in mice functionally lacking CSF-1 [117]. In a mouse model of breast cancer, defects of CSF-1 caused delays in the development of metastatic tumors [118].

Two representative mechanisms are known for the immunosuppressive activity of TAM. The first involves regulation through direct interactions with immunocompetent cells [119]. Here, TAM suppresses the activation of the immune mechanism by the direct transmission of negative signals to the immunocompetent cells [119]. It expresses immune checkpoint molecules, such as PD-L1 and CD80/CD86, which are recognized by PD-1 and CTLA-4, respectively, present on the CTL, resulting in inactivation of these CTLs [120]. Furthermore, TAM expresses HLA-G and HLA-E, which are suppressive MHC-1 molecules [121]. HLA-G is recognized by LIT-2 (leukocyte immunoglobulin-like receptor 2) expressed on CD4+ T cells, resulting in suppressed activation of these cells [120]. Similarly, HLA-E is recognized by NKG2 on natural killer cells, resulting in suppressed migration of these cells and suppressed secretion of IFN-γ [120]. The second mechanism involves the recruitment of Treg. TAM regulates immune responses by recruiting Treg into the tumor microenvironment [121]. TAMs are known to recruit natural Treg into tumor tissues by producing chemokines (CC-chemokine ligands: CCL3, CCL4, CCL5, CCL20, CCL22) [122]. Furthermore, by producing IL-10 and TGF-β, TAM activates the transcription factor Foxp3 of CD4+ T cells, possibly contributing to the induction of inducible Treg [121,123].

Other than these mechanisms, immune responses are regulated by chemokines, cytokines, etc. secreted from TAM and cancer cells. For example, chronic pathogen infections stimulate the continuous production of inflammatory cytokines (IFN-γ, TNF, IL-6, etc.) by macrophages, resulting in chronic inflammation [124]. Such a state of chronic inflammation may stimulate tumor progression. Furthermore, TAM induces cancer cell migration, infiltration, intravascular invasion and neovascularization needed for tumor growth, and this can lead to tumor metastasis [122]. For example, in breast cancer, CSF-1 secreted from cancer cells and vascular endothelial growth factor (VEGF) secreted from TAM mutually stimulate the secretion of these factors, resulting in the acceleration of tumor infiltration and intravascular invasion [125]. TAM is also involved in remodeling of the tumor microenvironment through the expression of proteases such as matrix metalloproteinase (MMP), cathepsin S, urokinase type plasminogen activation factor, and matrix remodeling enzymes, such as lysyl oxidase and SPARC [122]. These proteases cut the extracellular matrix and create free space around the tumor, thus stimulating the secretion of growth factors, such as heparin-binding epidermal growth factor (HB-EGF) and resulting in tumor infiltration and metastasis [122].

Macrophages also play an important role in tumor neovascularization. TAMs secrete neovascularization factors, such as VEGF, TNF, IL-1β, IL-8, PDGF, and FGF and are thus, involved in the stimulation of neovascularization around the tumor [126]. In the tumor microenvironment, a low oxygen supply (hypoxia) can arise from malnutrition, extracellular pH reduction, and insufficient blood flow. For cancer cells to survive, hypoxia needs to be avoided by inducing neovascularization and increasing blood flow to the tumor environment—the hypoxia inducible factor (HIF) plays an important role in this process. Excessive activation of HIF by TAM is often seen, resulting in the stimulation of VEGF expression and enhancement of neovascularization [127]. The molecular mechanism for this process has been increasingly elucidated. Recent studies have demonstrated that the lactic acid formed by cancer cell’s energy metabolism stabilizes TAM, resulting in the induction of VEGF expression [128].

The involvement of macrophages in tumor metastasis has also been observed in various studies. For example, in the foci of lung cancer metastasis, a small blood clot collecting cancer cells is first formed. Then, CCL2 is secreted from these cancer cells and recruits CCR2+Ly6c+ inflammatory monocytes, resulting in differentiation into Ly6c-metastasis associated macrophage (MAM) [129,130]. The monocytes and MAM recruited into the foci of metastasized tissue stimulates extravascular migration of cancer cells through VEGF expression. Moreover, cancer cell VECAM1 (CD106) is linked to MAM’s integrin α4 (VECAM1 counter-receptor), and MAM is involved in increases in metastatic cancer cell viability and tumor growth [129,130]. Likewise, defects in MAM inhibit tumor growth [125]. As illustrated below, TAM enables the tumor to escape immunity through diverse mechanisms (Figure 8).

## 9. Methods of Overcoming Tumor Immunoediting

### 9.1. Success of Immune Checkpoint Inhibitors in NSCLC Patients

The anti-CTLA-4 antibody is the first immune checkpoint inhibitor used for effective cancer treatment [131]. Its primary action points are the effective T cells and Tregs in the priming phase (Figure 9). Animal models and human samples reveal the induction of effector T cell activation and decrease the number of Treg in the regional lymph node and tumor following anti-CTLA-4 treatment [132,133].

The anti-PD-1 antibody is an immune checkpoint inhibitor that was clinically introduced after the anti-CTLA-4 antibody [134]. Its primary action is the activation of cancer-specific effective cells in the effector phase (Figure 9). Uncombined anti-PD-1 antibody treatment results in long-lasting anti-tumor activity against various types of cancer, including lung cancers [135]. In a clinical study concomitantly using anti-CTLA-4 antibody in patients with melanoma, treatments with this antibody yielded a high response rate of 52% [136].

PD-L1 is a ligand for PD-1 and is expressed primarily on APCs and tumor cells. It binds to PD-1 expressed on cytotoxic lymphocytes and suppresses the function of these lymphocytes. The expression of PD-L1 is correlated with a poor prognosis of many types of cancer, including melanomas, ovarian and pancreas cancer [137,138]. Anti-PD-L1 is also clinically available at present. The response rate reported to date is slightly lower than that of anti-PD-1 antibody, but a similar clinical efficacy has been reported [139].

Immune checkpoint inhibitors (ICIs) significantly advanced the treatment of NSCLC. The efficacy of ICIs targeting the PD-1/PD-L1 axis in NSCLCs has been demonstrated even in early-stage patients [140,141,142,143]. Long-term follow-up data demonstrate that immunotherapy has great potential for a long-term response. The results of KEYNOTE-024 and -042 support the monotherapy of pembrolizumab, immunoglobulin (Ig) G4-kappa monoclonal antibody against PD-1, as the first-line treatment for PD-L1-positive NSCLC patients [78,144]. Specifically, KEYNOTE-024 (randomized, open-label, phase III trial) showed that pembrolizumab significantly improved progression-free and overall survival (OS), as compared to platinum-based chemotherapy in patients with previously untreated NSCLC with PD-L1 high expression. Subsequent analysis of KEYNOTE-024 showed that the median OS was 30.0 months (95% CI, 18.3 months to not reached) with pembrolizumab and 14.2 months (95% CI, 9.8–19.0 months) with chemotherapy (hazard ratio, 0.63; 95% CI, 0.47–0.86). The first-line pembrolizumab monotherapy demonstrated an OS benefit over chemotherapy in previously untreated NSCLC patients [145].

Landmark analyses of pooled data from large clinical trials of nivolumab, IgG4 monoclonal antibody directed against PD-1, in previously treated NSCLC patients (CheckMate 017, 057, 063, and 003) showed that the four-year OS rate with nivolumab was 14% (95% CI 11–17) for all patients—19% for patients expressing PD-L1 and 11% for those without PD-L1 expression [146]. Thus, NSCLC patients treated with nivolumab achieved greater response durations compared with patients treated with docetaxel, resulting in a long-term survival benefit.

It has been reported that platinum-based chemotherapy can increase the sensitization of tumors to ICIs by increasing tumor-infiltrating immune cells [147]. KEYNOTE-021 was the first trial that successfully combined platinum-based chemotherapy and ICIs for previously untreated advanced NSCLC patients [148]. Recently, KEYNOTE-189 and -407 showed significantly improved patient survival rates following the administration of pembrolizumab combined with conventional cytotoxic chemotherapy. Intriguingly, the magnitude of survival benefits was similar irrespective of PD-L1 expression [149,150]. The randomized phase III clinical trial, IMpower150, reached their primary endpoint and demonstrated that enrolled patients gained better survival and fewer risks from the combination therapy: atezolizumab and chemotherapy, as compared to monotherapy with chemical agents. This trial concluded that the combination therapy resulted in better progression-free survival (PFS) rates than the monotherapy in the PD-L1 high expression and negative groups; in contrast, there were no significant differences of PFS between treatment groups in the PD-L1 low expression group, indicating that biomarkers for patient selection need to be further explored [151,152].

CTLA-4 is the first known immune checkpoint and is exclusively expressed on T cells [153]. Ipilimumab, IgG1 monoclonal anti-CTLA-4 antibody, was the earliest ICI approved by the FDA. The randomized phase III trial of ipilimumab combined with paclitaxel and carboplatin did not prolong OS of patients with advanced squamous NSCLC compared to the chemotherapy alone [154]. The combination of anti-PD-1 and anti-CTLA-4 ICIs has shown a manageable tolerability profile with anti-tumor activity and clinical benefit [155]. Based on this result, TMB has emerged as a potential biomarker of benefit, and a phase III trial was conducted to examine PFS with nivolumab plus ipilimumab versus chemotherapy among NSCLC patients with high TMB (≥10 mutations/Mb) [156]. The median PFS was 7.2 months (95% CI, 5.5–13.2 months) with nivolumab plus ipilimumab versus 5.5 months (95% CI, 4.4–5.8 months) with chemotherapy (hazard ratio, 0.58; 97.5% CI, 0.41–0.81).

As described above, ICIs have demonstrated durable survival benefits for advanced NSCLC patients. However, response rates to ICIs remain suboptimal, and many patients who initially respond to treatments ultimately developed progressive disease during follow-up monitoring [157,158]. Therefore, the next steps in developing novel immunotherapies are essential.

### 9.2. Treg-Targeting Treatment

Generally, effective anti-tumor immune reactions can be induced by gaining an upper hand in the quantitative and functional balance between the tumor-attacking effector T cells and Treg in a direction favoring the former. This can be achieved by monoclonal antibodies with various immunobiological activities (e.g., cytotoxic, blocking, and agonistic activities) to a variety of functional molecules. Prolonged systemic removal/decrease of Treg and suppression/attenuation of its function may induce autoimmunity as seen following anti-CTLA-4 antibody treatment [93,159]. However, short-term or local administration of this antibody, setting a target at a Treg subset, can induce strong anti-tumor immune responses without the side-effect of autoimmune diseases. The following may be listed as new regimens of cancer treatments targeting the Treg.

The use of anti-CCR4 antibodies enables the removal of CCR4-expressing Treg that invades the cancer tissue and could reinforce the induction of cancer-specific T cells [160]. The percentage of cancer antigen-specific cytotoxic lymphocytes has been shown to increase after treatment with an anti-CCR4 antibody from the pretreatment level in patients with adult T cell leukemia [109]. Simultaneously, treatment with the CCR4 antibody can prevent autoimmunity because the naïve Treg in blood and lymph tissue is preserved. Thus, as an intrinsic factor, anti-CCR4 antibody mogamulizumab is now under a phase I trial for its use in combination with nivolumab, etc., for the treatment of solid cancers (e.g., NSCLC).

Both OX-40 and GITR are expressed on activated T cells and are molecules belonging to the TNF superfamily [22]. OX-40 and GITR are constitutively expressed on Treg. These molecules suppress the function of Treg and activate the effector T cells. In mice, treatment with agonist antibodies to OX-40 and GITR result in a decrease of Treg and reinforced anti-tumor immune responses in the presence of various cancers [161,162]. In humans, the anti-tumor immune responses of OX-40 against malignant melanoma and renal cell carcinoma have already been confirmed in a phase I study [163]. Presently, clinical trials on OX-40 are underway (including its use in combination with anti-PD-L1 antibody) in patients with various types of cancer.

What is immunologically important for various methods of cancer treatment, is that effective anti-cancer immune responses may be suppressed if Treg is recruited and stimulated by the autoantigen, tumor antigen, or local inflammation released or induced because of the injury of the cancer cells. The removal/decrease of Treg or attenuation of Treg’s suppressive activity is expected to reinforce the efficacy of other anti-cancer immunotherapies [135]. For example, the anti-tumor activity of cancer vaccines administered independently is usually insufficient, but the efficacy of cancer vaccinations after Treg removal deserves evaluation. Furthermore, if the immune checkpoint therapy effectively activates the effector T cells, its anti-tumor activity may be reinforced by a decrease or removal of Treg [135]. Thus, advances in new anti-cancer immunotherapy targeting the Treg are promising.

### 9.3. TAM-Targeting Cancer Treatment

Because TAM regulates the development, progression, and metastasis of tumors, cancer treatment methods targeting the macrophages are now under development. For example, because the chemokine CCL5 of cancer cell origin is important in recruiting macrophages, a CCL5 receptor inhibitor was administered to mouse models of breast cancer [164]. This resulted in a decrease in the macrophages invading the tumor and a reduction of the tumor size. In addition, CCR2 inhibitors are now being developed to inhibit the CCL2-CCR2 interactions, which induce the migration of peripheral blood monocytes to the tumor tissue. When the CCR2 inhibitor was administered independently, the anti-tumor activity was limited. However, more potent anti-tumor activity was reported when it was used in combination with conventional chemotherapy [165].

M2 macrophages contribute to anti-inflammatory reactions and tumor growth, while M1 macrophages (proinflammatory macrophages) are involved in anti-tumor responses. Based on this view, an attempt is now being made to suppress tumorigenesis by causing polarization of TAM into M1 macrophages. For example, the administration of CpG and anti-IL-10 antibody to mice induced polarization of M2 macrophages into M1 macrophages, resulting in tumor cell death [166]. Furthermore, inhibition of SHIP1, an important phosphatase inducing M2 macrophages from M1 macrophages, has been shown to inhibit the polarization of M1 macrophages into M2 macrophages and cause an increase in M1 macrophage invasion, resulting in intensified anti-tumor responses [167]. In humans, close attention has been paid to the CSF-1R inhibitor/inhibitory antibody. This is now under clinical trials for development to remove TAM by inhibiting the signals of CSF-1 (a macrophage survival/differentiation factor). Drugs targeting the signal transduction pathway mediated by NF-κB or STAT3 and anti-D40 agonistic antibody, PI3Kγ inhibitors and histone deacetylase inhibitors are also expected to manifest anti-tumor activity through regulation of TAM’s immunosuppressive activity, particularly through stimulation of differentiation into M1-like macrophage and activation of anti-tumor immunity [112,168].

Treatment aimed at reactivation of macrophage phagocytosis is an approach targeting the modification of TAM using antibodies such as the CD47 inhibitory antibody. The antibody is expected to reinforce anti-tumor immunity by promoting antigen presentation triggered by phagocytosis. In a clinical study involving malignant lymphoma patients, approximately half of the patients responded to the treatment with this antibody when combined with an anti-CD20 antibody [169].

These drugs targeting the TAM are expected to improve the efficacy of existing chemo-, radio-, and immuno-therapies by activating anti-tumor immune responses and have the potential of expanded applicability to patients poorly responding to existing therapies.

### 9.4. Phagocytosis Checkpoint Inhibitors

The innate immune response is vital for activating the adaptive immune system through a process of cross-priming, wherein APCs mediate the processing and presentation of antigens to naïve T cells, leading their activation [170]. Integral to this bridging process between innate and adaptive immunity is the ability of APCs to engulf tumor cells through phagocytosis—a multistep cellular event, including recognition of the target cell, cellular engulfment, and lysosomal digestion, which are regulated by receptor–ligand interactions between the target cell and the APCs (Figure 10) [171]. Although non-cancerous cells can escape phagocytosis by expressing anti-phagocytosis molecules, tumor cells heavily depend on similar mechanisms to evade the immunosurveillance by APCs (Figure 10) [171,172]. Hence, the therapeutic use of phagocytosis checkpoints in tumor cells might open a new avenue for the development of immunotherapies to inhibit tumor immune escape. The CD47-signal-regulatory protein α (CD47-SIRPα) axis is the first identified tumor phagocytosis checkpoint (Figure 10) [173,174]. Inhibiting the CD47-SIRPα interaction promotes the phagocytosis of tumor cells, which can be achieved therapeutically using anti-CD47 or anti-SIRPα antibodies. In the clinical setting, CD47-SIRPα axis blocking agents are nowadays being evaluated in many clinical trials, either as monotherapies or in combination therapies (Table 2) [175]. In vitro, dual blockade of the PD-1-PD-L1 axis and CD47-SIRPα induced immune cell activation and suppressed tumor growth and metastases in a mouse xenograft model. The combination of the CD47-SIRPα axis and PD-1-PD-L1 axis blockade therapies could serve as next-generation immunotherapy to inhibit innate and adaptive immune checkpoints (Table 2) [176].

### 9.5. Cancer Vaccines

DCs are a diversified group of specialized APCs with main functions in the initiation and regulation of innate and adaptive immune responses [177]. In the cancer microenvironment, the anti-tumor functions of DCs are impaired. Reduced availability of FMS-like tyrosine kinase 3 ligand (FLT3L) in the microenvironment suppresses terminal differentiation of DCs [178], and cytokines, such as IL-6, IL-10, and TGFβ, can affect both in situ and bone marrow generation of DCs [179]. Tumors can prevent the infiltration of DCs by limiting the expression of DC chemo-attractants, such as CCL4, and avoid detection by suppressing the release of activating molecular cues [180]. DC maturation is prevented by soluble mediators, such as IL-10, TGFβ, IL-6, or VEGF, which interfere with activation signals [177,180]. The tumor microenvironment is rich with immunosuppressive factors that limit the immunostimulatory capacity of DCs.

Restoring normal DC function is a powerful weapon to combat tumors. Targeted delivery of tumor antigens and adjuvants to DCs can improve the anti-tumor activity of DCs. C-type lectin receptors show a diverse expression pattern on DCs and have been used as preferential target receptors, such as DEC205, CLEC9A, and CLEC4A4 [181,182,183]. Antibody-conjugated antigens with adjuvant outperformed non-conjugated antigens. The full-length New York esophageal squamous cell carcinoma 1 (NY-ESO-1) antigen fused to anti-DEC205 antibodies promotes additional CD8+ T cell activation compared to uncoupled NY- ESO-1 [184]. In a phase I clinical trial, treatment of cancer patients with cutaneously injected NY- ESO-1 coupled to anti-DEC205 with resiquimod (R-848) and/or poly-ICLC increased the production of antigen-specific antibodies and T cells, leading to partial clinical responses without toxicity [185]. Delivery of adjuvants and antigens to DCs in vivo by targeting DC-restricted receptors is expected to enhance efficacy and reduce the side effects of adjuvants (Figure 11).

Cancer vaccine therapy is aimed at rejecting cancer through the administration of cancer antigen-derived CTL epitope peptide into humans and reinforcement of the immune responses by the peptide-specific CTL (Figure 11) [177]. Cancer vaccines have gained significant popularity in recent years as a precision medicine approach to cancer immunotherapy. There are currently three therapeutic cancer vaccines approved by the FDA: Oncophage for kidney cancer, Sipuleucel-T for hormone-refractory prostate cancer, and BCG for early-stage bladder cancer [186]. In lung cancer, the self-adjuvanting mRNA cancer vaccine, CV9202, targeting NY-ESO-1, MAGEC1, MAGEC2, survivin, 5T4, and MUC1 is in clinical trial in combination with radiotherapy for advanced NSCLC patients [187,188] (Table 3). Another mRNA-derived cancer vaccine, CV9201, was well-tolerated with median PFS and OS of 5.0 months (95% CI 1.8–6.3 months) and 10.8 months (95% CI 8.1–16.7 months) from the first administration, respectively. Two- and three-year survival rates were 26.7% and 20.7%, respectively [188]. The combination of the cancer vaccine and anti-PD-1 ICI treatment is also in clinical trial for advanced NSCLC patients. Viagenpumatucel-L is a cell-based vaccine derived from a secreted heat shock protein, gp96-Ig, from an NSCLC cell line that induces antigen-specific T cell activation [189]. A phase II study (NCT02439450) is currently enrolling patients to investigate whether vaccination with viagenpumatucel-L combined with nivolumab or pembrolizumab is safe for advanced NSCLC patients stratified based on high or low levels of tumor-infiltrating lymphocytes (TILs).

The clinical use of DC vaccines for cancer has been broadly investigated, with many trials [177]. However, because the number of DCs is small, it is uncertain whether or not the information about the cancer antigen can be transmitted to the CTL reliably. For this reason, the DC vaccine therapy, involving the coculture of artificially cultivated DCs with the cancer antigens outside the body and subsequently returning these cells into the patient’s body, has recently been developed [177]. This approach is comprised of the isolation or in vitro generation and amplification of autologous DCs, their ex vivo manipulation, and reinfusion into patients (Figure 12).

These studies have been predominantly performed in patients with prostate cancer, melanoma, renal cell carcinoma or glioblastoma due to the immunogenic nature of these cancers. Notably, they demonstrated the clinical efficacy and safety of DC vaccination to induce anticancer immune responses [190,191,192]. The only clinically approved DC vaccine to date is sipuleucel-T (Provenge), which comprises autologous DCs loaded with a recombinant fusion protein composed of GM-CSF and prostatic acid phosphatase. It was shown to improve the median OS of patients with prostate cancer [193]. Recently, the efficacy of DC vaccines is expected to be further improved by tuning the ideal amount of antigen loading and prompting DC maturation.

It is the quality, not quantity, of tumor neoantigens that may best predict responses to immunotherapies and the likelihood of long-term survival among patients with cancer [194] (Table 2). A patient-specific, personal approach targeting multiple epitopes derived from tumor neoantigens appears to be the most plausible precision medicine approach. Neoepitopes encoded by tumor mutations are identified and estimated based on sequencing information from tumor and germline cells, expression validation by RNA sequencing and mass spectrometry, and prediction of peptide binding to HLA molecules [195]. Personal vaccines directed at neoantigens can induce robust immune responses in the peripheral blood and trafficking of neoantigen-specific T cells into tumors, leading to anti-tumor activity in preclinical models [196]. Personal cancer vaccine directed at tumor neoantigens enabled by recent sequencing techniques and big data analysis is promising and will likely reach the clinic in recent years.

### 9.6. Chimeric Antigen Receptor T Cell and Bispecific Antibodies for Lung Cancer

Adoptive immunotherapy with gene-engineered chimeric antigen receptor (CAR) T cells is a concept involving massive and reliable preparation of T cells with the capability of directly attacking the cancer cells through genetic engineering and administration of such cells to the patients (Figure 13). It has obtained clinical proof of concept in hematologic cancers [197]. CAR-T cells can recognize proteins, sugar chains, and lipids on the cell surface without being restricted by MHCs and can therefore also deal with the cancer cells from which HLA has been lost [198,199]. In particular, CD19 is an exemplary target antigen in B cell leukemia and lymphoma [200,201]. However, in solid tumors, it is difficult to design CAR-T cells, because no surface antigens as specific as CD19 have been identified [202]. Many clinical trials to investigate the effects of CAR-T cells in advanced lung cancer patients have been initiated, including tumor-associated antigens of *EGFR*, *HER2*, and *CEA* [203]. One clinical trial of *EGFR*-specific CAR-T cells for NSCLC (NCT01869166) had reported its preliminary results—45.5% (5/11) of advanced NSCLC patients achieved stable disease, and two achieved partial responses. Treatment-related adverse events were manageable, indicating its potentials for treating NSCLC [204]. However, there are still some challenges to overcome in CAR-T therapy for solid tumors. First, off-target effects are the primary causes of increased toxicity. The targeted tumor-associated antigen can be expressed in other tissues or organs than the lung, which may lead to unexpected treatment-related toxicity. Second, because solid tumor are highly heterogenous, the CAR-T cells which attack a single target molecule is less effective against cancer cells with reduced or lacking target molecule. Third, the tumor microenvironment suppresses the immunoactivity of CAR-T cells. Thus, while promising, CAR-T cell-based therapies will require some refinements, such as (i) identification of neoantigens with higher tumor specificity; (ii) development of next-generation CAR-T with elevated potential of gathering in the tumor tissue; (iii) regulation of immunosuppression in tumor microenvironment, to be used more broadly.

## 10. Future Perspective

For anti-cancer immunotherapies to be effective against more types of cancers in a larger number of patients, it is important to develop composite anti-cancer immunotherapy made of a combination of various treatment methods. The basic concept is combining multiple treatment strategies capable of correcting and reinforcing each step during the “cancer immune cycle” (Figure 1) [8]. The cancer immune cycle involves numerous immunocompetent cells/molecules and gene anomalies, and the steps developing disorders, often in combination, and their cause vary from one patient to another. To create effective anti-cancer immunotherapies, it is essential to assess the entirety of the immune status in each tumor and reactivate the cancer immune cycle through the rational combination of treatment methods so that anti-tumor immune reactions may be induced. For example, the combination of Treg suppressor and immune checkpoint inhibitor and the combination of immunocompetent cell activation promoter and immune checkpoint inhibitor are promising. For now, it is expected that the development of various composite anti-cancer immunotherapies based on the PD-1/PD-L1 blockade will be advanced and introduced clinically, further elevating the importance of immunotherapy in cancer treating strategies.

## 11. Conclusions

The cancer immunosurveillance and immunoediting mechanism hypotheses began with the proposal of the attractive concept “recognition of the immune system by cancer” and has stimulated discussion over the effective, but incomplete control of cancer by the immune system. Thus, the importance of neoantigens based on individual cancer mutations has begun attracting close attention. The extreme success of ICIs in advanced tumors, especially in lung cancer and melanoma, has elucidated how our immune system can be leveraged to eradicate cancers. ICIs have broken new ground in the cancer therapy field. However, there are still many patients suffering from advanced cancers with no hopeful treatment options. Moreover, a new challenge is that tumors can acquire resistance to ICIs. To overcome these problems, we will need to more deeply understand the fundamental mechanisms of cancer escape from the immune system and develop new treatment strategies. The fight against cancer to utilize the immune system has just begun and is still budding. We anticipate that the next generation of cancer immunotherapies will be one step closer to the eradication of cancer.

## Figures and Tables

**Figure 1 ijms-21-00597-f001:**
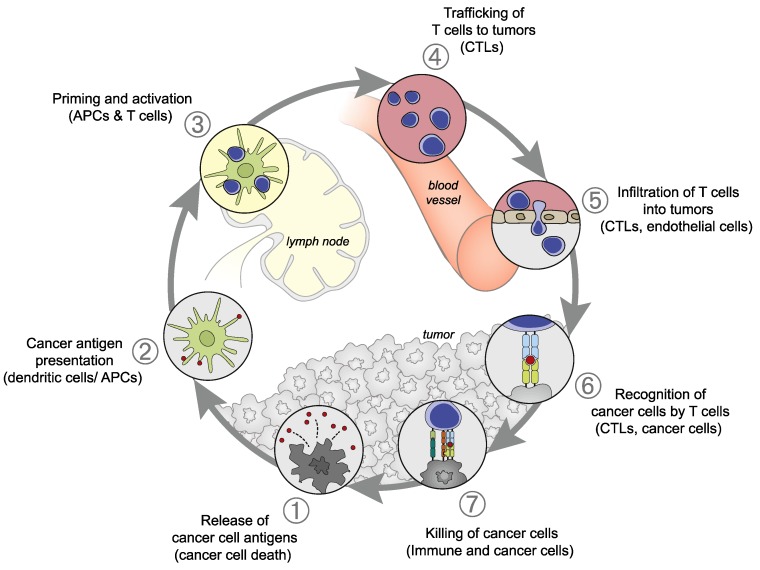
Cancer-immunity cycle: (1) Release of cancer antigens from tumor cells; (2) Presentation of cancer antigens on the major histocompatibility complex class (MHC) by antigen-presenting cells; (3) recognition of cancer antigens on the MHC by the T cell receptor, resulting in T cell activation; (4) Trafficking of activated T cells; (5) Infiltration into the tumor; (6) Recognition of cancer antigens on the MHC within the tumor; (7) Attack on tumor cells, resulting in tumor cell injury/death. This figure was adopted from Chen DS et al. [8].

**Figure 2 ijms-21-00597-f002:**
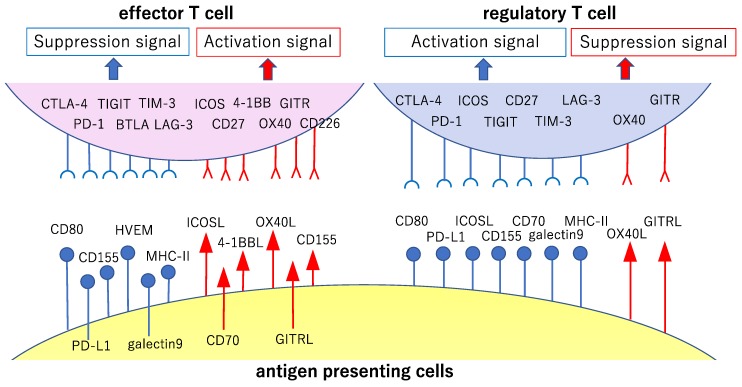
Major co-stimulatory and co-suppression molecules involved in the priming phase of cancer immunity. The cancer antigen presented by antigen-presenting cells in the lymph nodes is recognized via the T cell antigen receptor (TCR) by cancer-specific T cells (signal 1, main signal). Major co-stimulatory and co-suppression molecules producing auxiliary stimuli (signal 2, sub-signal) during this step and their ligands are shown here. The signals reinforcing the immune responses to cancer are shown in red, while those the attenuation of these responses are shown in blue.

**Figure 3 ijms-21-00597-f003:**
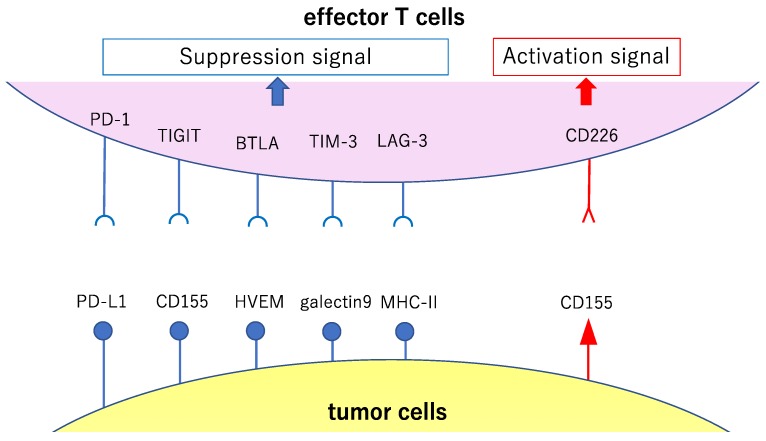
Major co-stimulatory and co-suppression molecules involved in the effector phase of cancer immunity. This figure shows major co-stimulatory and co-suppression molecules (involved in the attack against the cancer cells by activated and proliferated effector T cells after antigen presentation and recognition) and their ligands. The signals reinforcing the immune responses to cancer are shown in red, while the signals causing attenuation of such responses are shown in blue.

**Figure 4 ijms-21-00597-f004:**
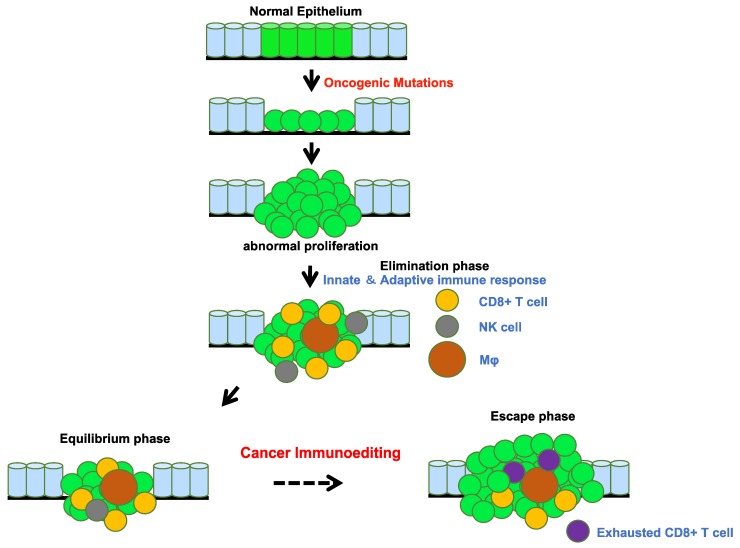
Tumor evolution and immunoediting. The accumulation of oncogenic mutations initiates tumor evolution. The immune surveillance system, including cytotoxic CD8 positive T cells, natural killer cells, and macrophages, recognize tumor-associated neoantigens presented by MHC molecules on antigen-presenting cells. The balance between signals from the tumor microenvironment and the immune system shifts during tumor elimination, equilibrium, and escape.

**Figure 5 ijms-21-00597-f005:**
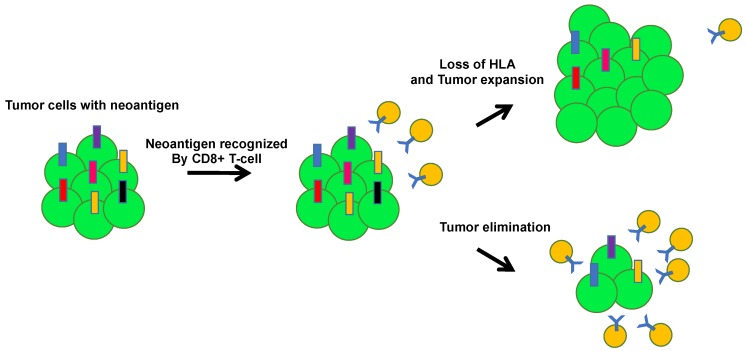
Tumor escape through human leukocyte antigen (HLA) loss. HLA loss allows tumor cells to escape the immune system. During tumor evolution, the accumulation of tumor neoantigens induces local immune infiltration. Tumor cells with HLA loss can be positively selected by avoiding CD8 T cell recognition.

**Figure 6 ijms-21-00597-f006:**
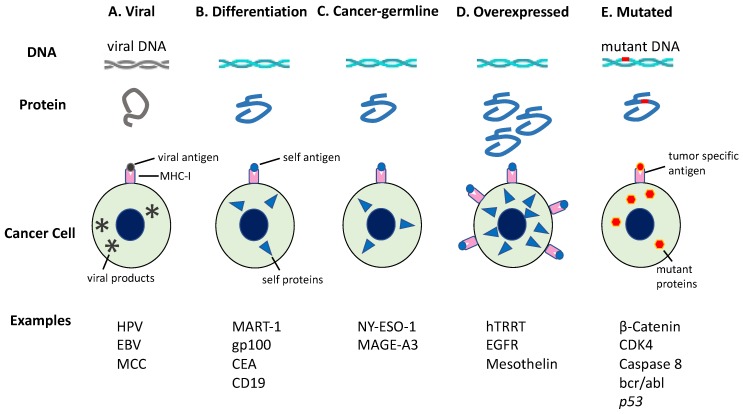
Tumor antigens recognized by immune cells. Tumor antigens are categorized based on the pattern of gene expression. The production of antigenic peptides by cancer cells is illustrated herein. (**A**) Viral antigens are only expressed in virus-infected cells; (**B**) Differentiation antigens are encoded by genes with tissue-specific expression; (**C**) Cancer-germline genes are expressed in tumors or germ cells because of whole-genome demethylation; (**D**) Some genes are overexpressed in tumors owing to increased transcription or gene amplification. The resulting peptides are upregulated on these tumors, but also show a low level of expression in some noncancerous tissues; (**E**) However, mutated genes may yield a mutant peptide (neoantigen), which is recognized as nonself by immune cells. CEA, carcinoembryonic Ag; EGFR, epidermal growth factor receptor; EBV, Epstein-Barr virus; HPV, human papillomavirus; hTERT, human telomerase reverse transcriptase; MAGE-A3, melanoma-associated antigen 3; MART-1, melanoma antigen recognized by T cells-1; MCC, Merkel cell carcinoma. This figure is adapted from our previous article [63].

**Figure 7 ijms-21-00597-f007:**
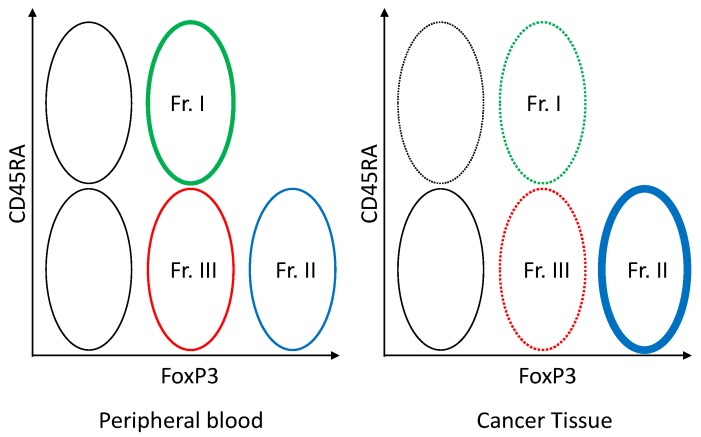
Foxp3+ T cell subset in the blood and tumor tissue. FoxP3^low^CD45RA+ naïve Treg (Fr. I) can be viewed as cells that have just been supplied from the thymus. If these naïve Treg are activated by the TCR signal, CD45RA becomes negative and FoxP3 is upregulated, yielding FoxP3^high^CD45RA- effector Treg (Fr.II). Effector Treg (Fr. II) expresses high levels of CCR4. Regarding FoxP3+ cells, the CD45RA negative FoxP3^low^ CD4+ T cell (Fr. III) which has no immunosuppressive activity is not Treg, but a helper T cell producing IFN-γ or IL-17 when activated.

**Figure 8 ijms-21-00597-f008:**
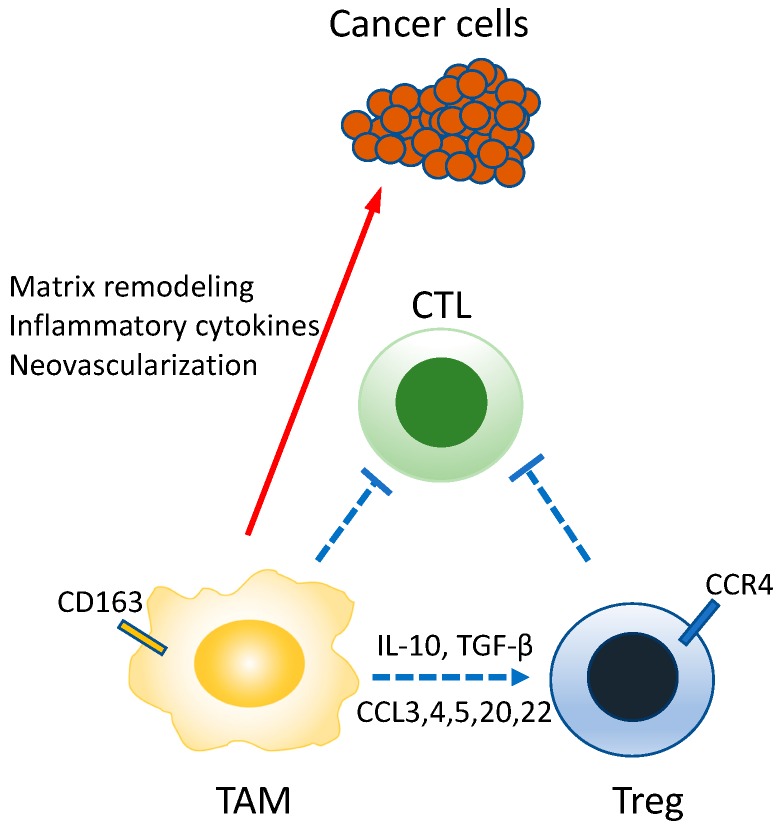
Roles of TAM in the tumor environment. TAM has the nature of M2-macrophage, directly suppressing the CTL, but also producing mediators involved in neovascularization, tumor infiltration/metastasis, and Treg migration/maintenance. The immunosuppressive activity is represented as a blue line and the tumor-stimulating activity as a red line.

**Figure 9 ijms-21-00597-f009:**
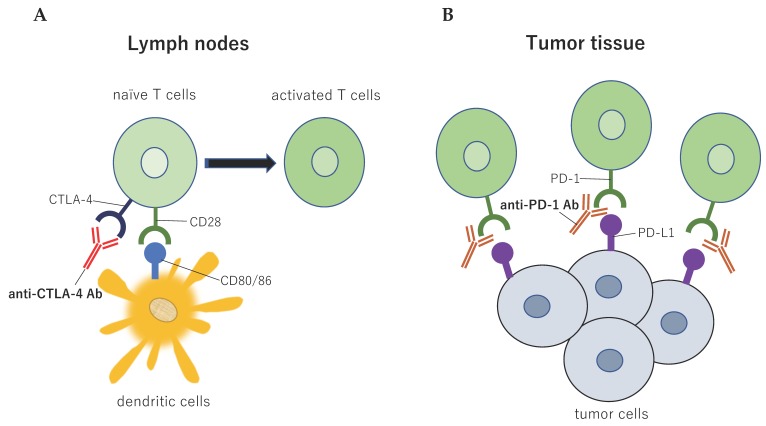
Immune checkpoint structure between tumor cells and immune cells. Tumor antigens are presented to T cells via the interaction between the major histocompatibility complex and T cell receptors, which is the primary signal for activating T cells. Several sub-signals function as negative modulators of the immune response at different molecular checkpoints. (**A**) The cytotoxic T-lymphocyte associated protein 4 (CTLA-4) is induced in T cells at the time of the priming phase. CTLA-4 is expressed on the cell surface in proportion to the amount of antigen stimulation, and then CTLA-4 binds to CD80/86 with greater affinity than CD28, leading to specific T cell inactivation; (**B**) The binding of programmed cell death ligand 1 (PD-L1) to PD-1 prevents T cells from killing tumor cells in the body. Blocking the binding of PD-L1 to PD-1 with an immune checkpoint inhibitor allows the T cells to kill the tumor cells. This figure is adapted from our previous article [63].

**Figure 10 ijms-21-00597-f010:**
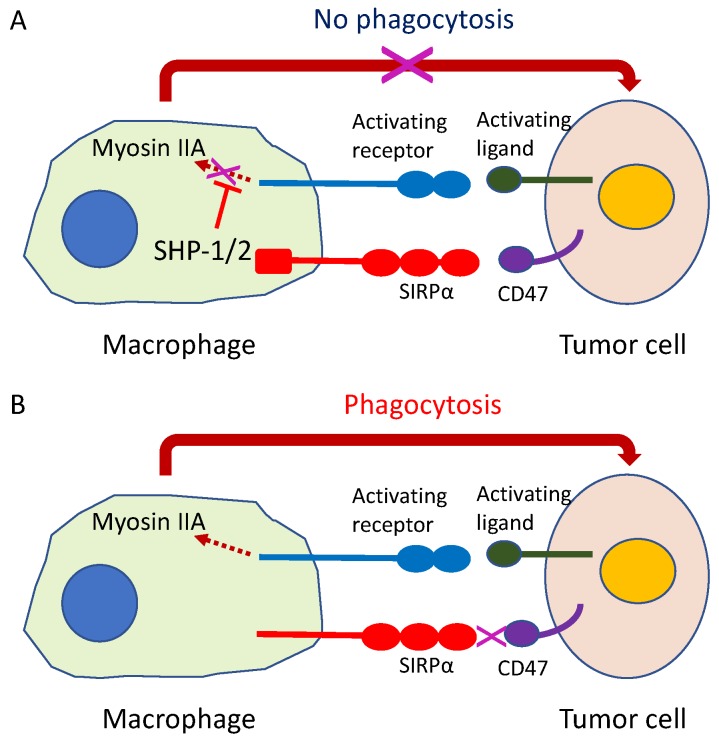
Innate immune checkpoint. The intercellular signal CD47-SIRPα system, formed by the signal regulating protein (SIRP) α and membranous protein CD47 located in the cell membrane of macrophages with phagocytosing capability functions as an innate immune checkpoint. (**A**) If the SIRPα on macrophage binds to CD47 on cancer cells, the cancer cells release a phagocytosis avoidance signal for macrophages to escape phagocytosis; (**B**) If this checkpoint function is inhibited, the macrophage begins tumor cell phagocytosis.

**Figure 11 ijms-21-00597-f011:**
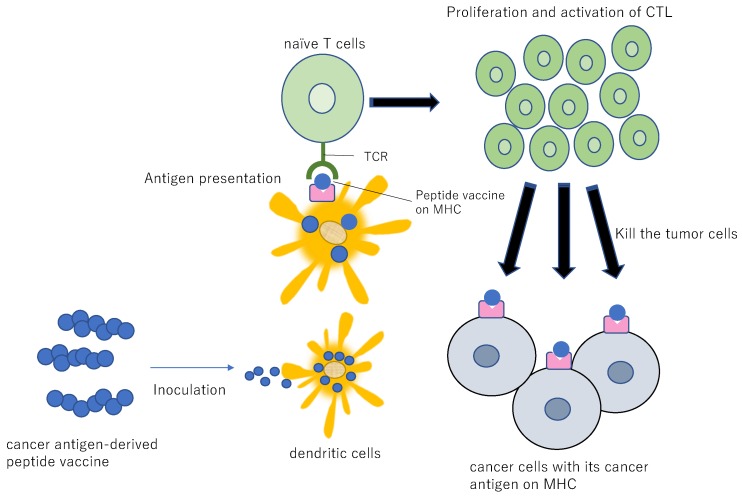
Diagram for peptide vaccine. If a peptide vaccine originating from a cancer antigen is inoculated into a patient, the peptide is taken up by the DCs, resulting in the presentation of the antigen to the CTL. This antigenic stimulus induces proliferation and activation of CTL, resulting in specific injury/killing of the cancer cells presenting the cancer antigen peptide.

**Figure 12 ijms-21-00597-f012:**
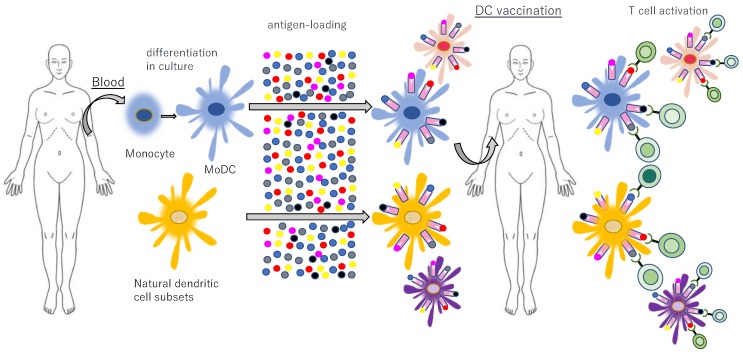
Adoptive transfer of autologous, antigen-loaded and activated DCs. Natural DC subsets and blood monocytes are isolated from blood and monocytes are cultured in vitro, differentiating into monocyte-derived DCs (MoDCs). After ex vivo activation and antigen loading, autologous DCs are reinfused into the patient to induce and activate antigen-specific T cells with strong efficacy and minimal side effects.

**Figure 13 ijms-21-00597-f013:**
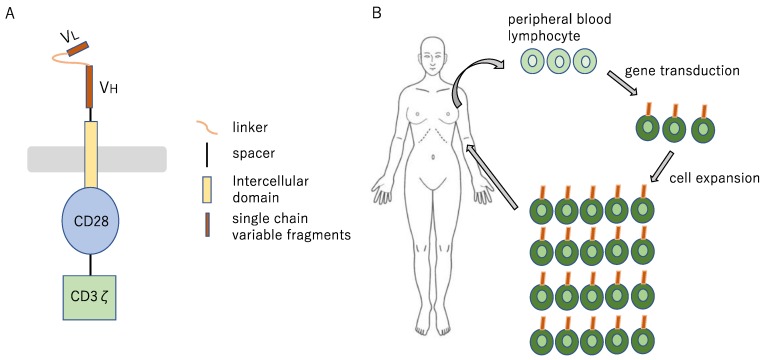
Cell therapy concept diagram (**A**) The Fab portion, corresponding to the surface antigen of the target tumor cell, is utilized in the form of single-chain antibody, and the signal transduction site (CD3ζ) of the T cell receptor and the CD28 (a co-stimulatory molecule for T cells) are fused to it, to yield CAR; (**B**) CAR is introduced by means of gene transfer with a retrovirus vector into T cell, to yield CAR-T cell. The T cell collected from the patient is genetically modified ex vivo, to yield CAR-T cell, which is infused into the patient after expanded culture.

**Table 1 ijms-21-00597-t001:** Co-stimulatory and co-suppression molecules involved in cancer immunity and their activities.

Co-Stimulatory Molecule	Ligand	Expression on Teff	Effects on Teff Function	Expression on Treg	Effects on Treg Function
CTLA-4	CD80, CD86	induced by stimulation	↓	constitutively expressed	↑
PD-1	PD-L1	induced by stimulation	↓	constitutively expressed	↑
ICOS	ICOSL	induced by stimulation	↑	constitutively expressed	↑
TIGIT	CD155	induced by stimulation	↓	constitutively expressed	↑
CD27	CD70	constitutively expressed	↑	constitutively expressed	↑
4-1BB	4-1BBL	induced by stimulation	↑	-	-
OX40	OX40L	induced by stimulation	↑	constitutively expressed	↓
GITR	GITRL	induced by stimulation	↑	constitutively expressed	↓
CD226	CD155	constitutively expressed	↑	-	-
BTLA	HVEM	constitutively expressed	↓	-	-
TIM-3	galectin-9	induced by stimulation	↓	constitutively expressed	↑
LAG-3	MHC-II	induced by stimulation	↓	induced by stimulation	↑

Red arrows indicate the signals reinforcing the immune responses to cancer, while blue arrows indicate the signals causing attenuation of such responses. Teff: effector T cell. Treg: regulatory T cell.

**Table 2 ijms-21-00597-t002:** Clinical trials investigating phagocytosis checkpoint blockades.

Clinical Trials, Gov Identifier	Phase	Intervention	Trial Design	Cancer Type	Primary End Points
**Monotherapy Trials**					
NCT03763149	I	Anti-CD47 antibody (IBI188)	Dose escalation	advanced malignancies and lymphoma	safety and tolerability
NCT02678338	I	Anti-CD47 antibody (Hu5F9-G4)	Dose escalation	hematological malignancies	tolerability
NCT02216409	I	Anti-CD47 antibody (Hu5F9-G4)	Dose escalation	solid tumors	safety and tolerability
NCT03834948	I	Anti-CD47 antibody (AO-176)	Dose escalation Dose expansion	solid tumors	safety and tolerability
NCT03013218	I	high-affinity SIRPα fusion protein (ALX148)	Dose escalation	solid tumors and lymphoma	dose-limiting toxicity
NCT03512340	I	Anti-CD47 antibody (SRF231)	Dose escalation Dose expansion	solid tumors hematological malignancies	safety and tolerability
**Combination Trials**					
NCT02367196	I	Anti-CD47 antibody (CC-90002) alone or in combination with rituximab	Dose escalation	solid tumors and hematological malignancies	safety and tolerability
NCT02663518	I	SIRPαFc (TTI-621) alone or in combination with rituximab or nivolumab	Dose escalation	relapsed/refractory hematological and solid malignancies	safety and tolerability
NCT02890368	I	SIRPαFc (TTI-621) alone or in combination with an anti-PD-1/PD-L1 agent, pegylated IFNα2a, T-VEC or radiation	non-randomized parallel assignment	solid tumors and mycosis fungoides	optimal delivery regimen
NCT03248479	Ib	Anti-CD47 antibody (Hu5F9-G4) alone or in combination with azactidine	non-randomized	AML and MDS	safety and tolerability
NCT02953509	Ib/II	Anti-CD47 antibody (Hu5F9-G4) in combination with rituximab	single-arm, non-randomized	refractory/relapsed non-Hodgkin lymphoma	safety and tolerability
NCT02953782	I/II	Anti-CD47 antibody (Hu5F9-G4) in combination with cetuximab	single-arm, non-randomized	solid tumors and CRC	safety and tolerability

**Table 3 ijms-21-00597-t003:** Tumor-associated antigens used to activate DCs.

Tumor Antigens	Proteins	Specificity	Advantages	Disadvantages
Differentiation antigens	MART1, GP100, PAP, CEA	Low	high prevalence, cheap off-the-shelf products, allow conjugation	high probability of nonspecificity and side effects
Overexpressed antigens	WT1, MUC1, ERBB2	Low	high prevalence, cheap off-the-shelf products, allow conjugation	high probability of nonspecificity and side effects
Viral antigens	HPV-, EBV-derived proteins	High	very specific, allow conjugation	limited prevalence of virus-associated tumors
Cancer-Germline antigens	NY-ESO-1, MAGE, GAGE, BAGE	High	specific, cheap off-the-shelf products allow conjugation	not exclusive to cancer
Mutated neoantigens	mutated tumor neoantigens	Highest	very specific, allow conjugation	expensive, labor- and technology intensive
Whole tumor antigens	lysate of cancer material	Variable	no need for neoantigen identification, contain additional DC-activating factors	limited cancer material, uncontrolled material

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
