# Peer review of "Immunosurveillance and Immunoediting of Lung Cancer: Current Perspectives and Challenges"

_ijms, 2020, doi:10.3390/ijms21020597_

Round 1
Reviewer 1 Report
The review entitled “Immunosurveillance and Immunoediting of Lung Cancer: Current Perspectives and Challenges” by Kunimasa et al represents a brief and precise summary of the state of the art of “cancer immunoediting”. The paragraphs are well described and sub-divided and the literature is also very rich. In my opinion, the review takes up general characteristics and in the title as in the content it is not necessary to focus on lung cancer. The review could instead report the latest successes in “recognation of the immune system by cancer” in cancer in general, and represent an in-depth and interesting reading on this topic.
Line 154: Giving recent success in cancer, I suggest to include a brief update for all cancer and not only in lung cancer
Line 161. I suggest to include notions about HLA loss and relapse
Line 228 I suggest to include reference on tumor clonal evolution and metastasis, regarding NGS studies.
Line 239. I suggest to include references regarding NGS approaches for neoantigen identification. I will describe the importance of fusion neoantigen.
Line 297. I will add reference to this sentence
Line 317 I will add reference to this sentences as PMID:31088845
Line 335 Regulatory T cells shuld be wrote as Treg in the test, after the first time mentioned
Author Response
Reviewer 1:
Line 154: Giving recent success in cancer, I suggest to include a brief update for all cancer and not only in lung cancer
Response: We added some descriptions, according to the reviewer’s suggestion.
Line 161. I suggest to include notions about HLA loss and relapse
Response: We added some descriptions, according to the reviewer’s suggestion.
Line 228 I suggest to include reference on tumor clonal evolution and metastasis, regarding NGS studies.
Response: We added some references to the last paragraph in section 5, as the reviewer suggested.
Line 239. I suggest to include references regarding NGS approaches for neoantigen identification. I will describe the importance of fusion neoantigen.
Response: We added several references to the explanation of neoantigen identification. Also, we added some descriptions about the fusion neoantigen, as the reviewer suggested.
Line 297. I will add reference to this sentence
Response: We added the references to the sentence.
Line 317 I will add reference to this sentences as PMID:31088845
Response: We added the references to the sentence.
Line 335 Regulatory T cells shuld be wrote as Treg in the test, after the first time mentioned
Response: We rephrased the abbreviation, as the reviewer suggested.
Thank you very much for your thoughtful comments. Please note that modified sections are highlighted in light blue in the revised manuscript.
Reviewer 2 Report
This was an extremely well written review. I recommend accepting as is, with no further edits.
Author Response
Reviewer 2:
This was an extremely well written review. I recommend accepting as is, with no further edits.
Response:
Thank you very much for your glowing comment.
Round 2
Reviewer 1 Report
I find this review to be well structured and rich in literature.Reading it is pleasant and interesting.